# The Limits and Avoidance of Biases in Metagenomic Analyses of Human Fecal Microbiota

**DOI:** 10.3390/microorganisms8121954

**Published:** 2020-12-09

**Authors:** Emma Bergsten, Denis Mestivier, Iradj Sobhani

**Affiliations:** 1EA7375 (EC2M3 Research Team), Université Paris Est, 94010 Créteil, France; emma.bergsten@u-pec.fr (E.B.); denis.mestivier@u-pec.fr (D.M.); 2Bioinformatics Core Facility, Institut Mondor de Recherche Biomédicale, UMR 955—Institut National de la Santé et de la Recherche Médicale—UPEC, 94010 Créteil, France; 3Service de Gastroenterologie, Hôpital Henri Mondor, Assistance Publique-Hôpitaux de Paris, 94010 Créteil, France

**Keywords:** metagenomic, 16S RNA, pipeline, biases, fecal microbiota

## Abstract

An increasing body of evidence highlights the role of fecal microbiota in various human diseases. However, more than two-thirds of fecal bacteria cannot be cultivated by routine laboratory techniques. Thus, physicians and scientists use DNA sequencing and statistical tools to identify associations between bacterial subgroup abundances and disease. However, discrepancies between studies weaken these results. In the present study, we focus on biases that might account for these discrepancies. First, three different DNA extraction methods (G’NOME, QIAGEN, and PROMEGA) were compared with regard to their efficiency, i.e., the quality and quantity of DNA recovered from feces of 10 healthy volunteers. Then, the impact of the DNA extraction method on the bacteria identification and quantification was evaluated using our published cohort of sample subjected to both 16S rRNA sequencing and whole metagenome sequencing (WMS). WMS taxonomical assignation employed the universal marker genes profiler mOTU-v2, which is considered the gold standard. The three standard pipelines for 16S RNA analysis (MALT and MEGAN6, QIIME1, and DADA2) were applied for comparison. Taken together, our results indicate that the G’NOME-based method was optimal in terms of quantity and quality of DNA extracts. 16S rRNA sequence-based identification of abundant bacteria genera showed acceptable congruence with WMS sequencing, with the DADA2 pipeline yielding the highest congruent levels. However, for low abundance genera (<0.5% of the total abundance) two pipelines and/or validation by quantitative polymerase chain reaction (qPCR) or WMS are required. Hence, 16S rRNA sequencing for bacteria identification and quantification in clinical and translational studies should be limited to diagnostic purposes in well-characterized and abundant genera. Additional techniques are warranted for low abundant genera, such as WMS, qPCR, or the use of two bio-informatics pipelines.

## 1. Introduction

The recent development of sequencing technology including the 16S rRNA gene and the whole metagenome sequencing (WMS) allowed for the study of unculturable bacteria. These technological tools have been used to characterize the comprehensive gut microbial communities in order to identify bacteria involved in health and disease [1]. Consequently, 16S rRNA gene sequencing is now widely used for the quantification of microbial communities in clinical and translational metagenomic trials. Part of the analysis involves the translation of bacterial sequences into taxonomic profiles and the estimation of their relative abundance in the community. This process is widely used to characterize changes in bacterial communities, also known as dysbiosis, which has been associated with a multitude of diseases including obesity [2], diabetes [3], colorectal cancer (CRC) [4,5,6], immune disorders [7], and inflammatory bowel disease [8]. Consequently, the characterization of the intestinal microbiota by 16S RNA-based analysis, has become an alternative to culture. Thus, growing numbers of investigators apply this process for microbiota characterization. However, it has been noted that results vary greatly between different research groups and publication. For instance, in the field of CRC, various studies have reported results with noticeable discrepancies [9,10,11,12]. Although geographic and lifestyle differences have been suggested as explanations, these discrepancies could also result from the various methodologies used in these studies. Therefore, the differences in the metagenomic and meta-analyses applied make the direct comparison of diseased patients and their controls as well as the comparison between studies difficult [13]. Briefly, varying DNA-extraction protocols, sequencing technologies, and the bioinformatic processing pipeline, which are applied for taxonomic assignation and quantification, could introduce analytical bias [14].

In the present study, we report DNA extraction methods, gene sequencing and bacteria-abundance profiling that are based on the 16S rRNA gene. Then, main algorithms for 16S rRNA gene taxonomical assignments are discussed using data from WMS outputs from the same samples.

## 2. Materials and Methods

### 2.1. Subjects

Fresh fecal samples from ten male healthy volunteers (mean age of 27.10 years old) with a BMI of 25.39 ± 1.04) (mean ± standard error of the mean (SEM)), were collected and submitted to DNA extraction by three different methods (G’NOME, PROMEGA, and QIAGEN).

A second dataset (referred to as the “French cohort”) has been used from our published work (at Mondor Hospital, Creteil, France—Prof. I Sobhani) and was sequenced in 2014 at the Genomics Core Facility, European Molecular Biology Laboratory, Heidelberg [9]. For this work, we used the 50 control sub-group from this cohort for which 16S as well as WMS had been performed.

### 2.2. DNA Extraction According to the G’NOME Protocol

The modified G’NOME DNA isolation kit^®^ (MP Biomedicals, La Jolla, CA, USA) as described by Furet et al. [15], was used according to the manufacturer’s instructions. In brief, fecal samples (150 mg) were thawed and homogenized by vortexing in 600 µL of the supplied cell suspension solution. Then, 100 µL of cell lysis denaturing solution was added, and the samples were incubated at 55 °C for 30 min at agitation (900 rpm). Then, 25 µL of Protease Mix was added, and the mixture was incubated at 55 °C for 120 min, during which it was agitated at 900 rpm. Then, 750 µL of 0.1-mm-diameter glass beads were added, and samples were subjected to the Beadbeater (Biospec, Bartlesville, OK, USA) for several rounds at 30 Hz frequency for a total of 10 min. Following this, 15 mg of polyvinylpolypyrrolidone (PVPP) (Sigma-Aldrich, Saint-Louis, MO, USA) was added. Samples were vortexed and centrifuged at 20,000× *g* for three minutes, and the supernatant was recovered. The remaining pellet was washed with 400 µL of TENP (50mM Tris [pH 8], 20 mM EDTA [pH 8], 100 mM NaCl, 1% PVPP) and centrifuged at 20,000× *g* for three minutes. The washing step was repeated twice. Nucleic acids were precipitated by the addition of one volume of isopropanol and centrifugation at 20,000× *g* for ten minutes. The pellet was resuspended in 400 µL of molecular water and 100 µL of the salt-out mixture and incubated at 4 °C for ten minutes. Samples were spun for ten minutes at maximum speed, and the supernatant was recovered. DNA was precipitated with 100% ethanol (1:2 *v*/*v*) at room temperature for five minutes, which was followed by centrifugation at 16,000× *g* for five minutes. The DNA was resuspended in 100 µL of Tris-EDTA (TE) buffer (10mM Tris pH 7.5, 1mM EDTA) and incubated overnight at 4 °C. As a final step, 50 µL of RNase Mix (MWG-Biotech AG, Ebersberg, Germany) was added and the DNA was incubated at 55 °C for 30 min at 300 rpm. DNA solutions were stored at −20 °C until further used.

### 2.3. DNA Extraction Using the PROMEGA Kit

The “PROMEGA protocol” was applied according to the Wizard Genomic DNA purification kit^®^ (Promega, Madison, WI, USA). It was modified as previously described [16]. In brief, aliquots (150 mg) of stool samples were thawed and homogenized in EDTA 50mM, 1:10 *v*/*v*. After two minutes of centrifugation at 14,000× *g*, bacterial pellets were resuspended in 480 µL of 50 mM EDTA, 100 µL of 50 mg/mL lysozyme (MP Biomedical, La Jolla, CA, USA), and 20 µL of 5 U/µL mutanolysin (Sigma-Aldrich, Saint-Louis, MO, USA) and incubated for one hour at 37 °C. The bacterial pellets were recovered after two minutes of centrifugation at 14,000× *g*, and the DNA was extracted by the Wizard Genomic DNA purification kit, which was used in accordance with the manufacturer’s instructions. The purified DNA was suspended in 150 µL of TE buffer and then stored at −20 °C until use.

### 2.4. DNA Extraction Using the QIAGEN Kit

The “QIAGEN protocol” was applied according to the manufacturer’s instructions for a QIAamp DNA Stool mini kit^®^ (Qiagen, Hilden, Germany). Fecal samples (150 mg) were collected in 2 mL buffer ASL and submitted to continuous mild vibration for one minute or until homogenization. All steps were carried out at room temperature (15–25 °C). Fecal lysates were heated for five minutes at 70 °C. Then, temperature was increased up to 95 °C to allow lysis of Gram-positive bacteria. Inhibitex tablet was added to the supernatant, which was continuously agitated until the tablet was completely dissolved. When the pellet feces particles and inhibitors were bound to the Inhibitex matrix, the sample was then centrifuged at 14,000 rpm for three minutes. An amount of 200 μL of the supernatant was then recovered. Then, 15 μL of proteinase K was added, incubated at 70 °C for ten minutes, and briefly centrifuged before the addition of 200 μL of ethanol (96–100%). The lysate was then homogenized and put in a QIAamp spin column. The QIAamp spin column was centrifuged (14,000× *g* for 2 min) Buffer AW1 and buffer AW2 were used according to the manufacturer’s instructions to yield DNA amounts of 15 to 60 μg (75–300 ng/μL), depending on the individual stool sample. DNA solutions were stored at −20 °C until use.

### 2.5. Quantification and Quality Control of Genomic DNA

Spectrophotometer, qubit measurement, and gel electrophoresis: the genomic DNA concentration was defined by fluorometric quantitation, using a Qubit^®^ dsDNA BR assay kit and a Qubit^®^ 2.0 fluorometer (Invitrogen, Carlsbad, CA, USA). DNA quality was measured by Nano drop^®^ (Thermo Fisher Scientific, Waltham, MA, USA), according to the 260/280 OD absorbance ratio. The integrity of DNA was assessed by electrophoresis gel: 10 µL of each extracted DNA (1 ng/µL) was analyzed on 1% agarose gel.

Oligonucleotide primers and probes used for specific bacteria selection: Quantitative Polymerase Chain Reaction (qPCR) using TaqMan was performed for total bacteria population (all bacteria) [17], for the dominant bacterial group *Bacteroides*/*Prevotella* [15,18], and for the genus *Bifidobacterium* [15]. qPCR using SYBR Green was performed for the bacterial species *E. coli* [19]. The primers and probes that were used in this study (Appendix A) were synthesized by Applied Biosystems.

qPCR was performed to quantify bacterial DNA targets and to assess PCR inhibition effects in the 10 healthy volunteers stool samples, according to three different DNA extraction protocols. Briefly, qPCR reactions were carried out in 96-well plates with TaqMan Universal PCR 2X Master Mix^®^ (Applied Biosystems) or with SYBR Green PCR 2X Master Mix^®^ (Applied Biosystems, Foster City, CA, USA). Each reaction was run in duplicate on a final volume of 25 µL with 0.2 mM final concentration of each primer, 0.25 mM final concentration of each probe, and 10 µL of appropriate dilutions of DNA samples with 0.2 μM final concentration of each primer, 0.25 μM final concentration of each probe, and 10 μL of appropriate dilutions of DNA samples, as previously described [15]. TaqMan and SYBR Green amplifications were carried out using the following ramping profiles, respectively: one cycle at 95 °C for ten minutes, followed by 48 cycles of 95 °C for 15 s, and then 60 °C for one minute, or one cycle at 95 °C for 15 s, followed by 48 cycles of 95 °C for 15 s, and then 60 °C for one minute. A melting step was also added to improve amplification specificity: 95 °C for 15 s, 60 °C for one minute, and +0.3 °C until 95 °C for 15 s.

The bacterial genomic DNA extracted from the feces of ten patients, using the three DNA extraction methods, was used for qPCR. For each patient, standard curves of all bacteria were generated from serial dilutions of a known quantity of genomic DNA (10 ng, 1 ng, 100 pg, 10 pg, 1 pg, and 0.1 pg). These were used to generate linear regression lines by plotting threshold cycles (Ct) versus initial bacterial DNA quantity. For each bacterial species or genus, results were presented as a logarithm of colony-forming units (CFUs) per gram of stool.

### 2.6. 16S rRNA and Whole Metagenome Sequencing

Fecal DNA of the 10 healthy volunteers was subjected to metagenome sequencing of the conserved 16S V3–V4 region. The following universal 16S rRNA primers were used for the PCR reaction: V3F (TACGGRAGGCAGCAG) and V4R (GGACTACCAGGGTATCTAAT) to target the V3-V4 region, which gives the highest specificity of targeted sequences [15,17]. Amplifications were carried out using the following ramping profile: 1 cycle at 95 °C for 10 min, followed by 40 cycles of 95 °C for 30 s, 60 °C for 1 min. For SYBR-Green^®^ amplifications, a melting step was added to improve amplification specificity. The amplicons were purified, quantified, and pooled at equimolar concentration. Sequencing was performed on the Illumina MiSeq platform. The “French cohort” stool samples had been submitted to the G’NOME kit for DNA extraction (see [9] for details). We selected the control population of the cohort (n = 50) on whom 16S rRNA and Shotgnum Metagenomics sequencing was performed.

### 2.7. Bionformatics Analysis

The “French cohort” were downloaded from the European Nucleotide Archive (ENA) database (http://www.ebi.ac.uk/ena) under the access number ERP005534. Amplicon and WMS data (paired-end reads) were quality checked using the fastQC software (version 0.41) and quality-controlled using the Trimmomatic software (version 0.35, minimum base quality score of 30, using a sliding window of 5bp).

For 16S datasets, quality checked paired-end reads were merged using ExpressionAnalysis [20]. They were also confirmed by using FLASH2 (version 2.2.0) [21].

We selected three representatives algorithms for the 16S rRNA dataset analysis of taxonomical assignation: MALT (v0.4.1, using recommended parameters for 16S analysis: “-*mif*” and “*SemiGlobal*” alignment) and MEGAN6 (Community edition, v6.17) for the comparative approach [22], QIIME1 for the OTU/clustering approach [23], and DADA2 [24] for the statistical inference of amplicon sequence variants (ASV)-based approach. For each pipeline, we used default parameters and the SILVA rRNA database, version 123 [25]. QIIME1 is now outdated, but this version has been largely used in literature. Consequently, it is worthy to provide the community information concerning its reliability when one wants to refer to previous published results.

For the WMS dataset, we used the mOTU-v2 pipeline [26]. Paired-end reads had already been processed, so no pre-processing was performed here.

It is widely accepted that 16S rDNA should not be used below the genus level [27]. We summarized counts to the genus level with in-house scripts that used the R software (version 3.6.1) for 16S rRNA and WMS taxonomical assignations. After summation at the genus level, counts were normalized relative to the total number of reads in the sample. OTU, taxon, AVS, or mOTUs that were above the genus levels but without an official NCBI taxonomy ID (taxid) (see below) were not considered in further comparisons. OTU tables and scripts are made available: https://github.com/dmestivier/Microbiota-16S-WMS.

### 2.8. Genus Name Correction

Even when we used the same 16S rRNA database by using the three 16S pipelines, we still noticed a few discrepancies between genus names, some of which were associated with the same NCBI taxid. We also found differences in genus names between the 16S and mOTU-v2 pipelines. Examples include “*g_Candidatus Soleaferrea*”/“*g_Candidatus_Soleaferrea*” (space or underscore separator), “*g_Chroococcidiopsis*”/“*g__Chroococcidiopsis_CC1*,” “*g__Lachnoclostridium*”/“*g_Lachnoclostridium_5*,” and “*g_Coriobacteriaceae_UCG*”/“*g__Coriobacteriaceae-UCG*” (no official NCBI taxonomy with “_CC1/_UCG/…” suffixes). Another example is “*g_Escherichia*” (MALT)/“*g_Escherichia-Shigella*” (QIIME1)/“*g__Escherichia/Shigella*” (DADA2). Notably, MALT also has a “*g_Shigella*” genus, which was not merged into the “*g_Escherichia*” genus because it is an official NCBI taxonomy genus name. Some genera were also characterized as “unknown” (e.g., “g_unknown Clostridiales”) or “uncultured” (e.g., “g_uncultured-bacterium”). These discrepancies impeded genus summation and abundance estimation, as well as their use for concordance between pipelines without filtering and correction. We manually corrected genus names to the official NCBI taxonomy name and filtered out “*unknown*” and “*uncultured*” genera.

### 2.9. Statistical Analysis

We used the shiny application for metagenomic analysis server (SHAMAN, [28]) for graphical visualizations, comparisons using Principal coordinate analysis, and differential abundance analysis (based on the DESeq2 package) [29,30]. Other figures were generated using in-house scripts with R. Heatmaps were generated using the pheatmap package (Version 1.0.12), Venn diagrams were produced using the VennDiagram package (Version 1.6.20) and intra-class correlation (ICC) analyses were performed using the psych package (Version 1.9.12.31).

### 2.10. Data Availability

OTU tables and Scripts are provided in GitHub: https://github.com/dmestivier/Microbiota-16S-WMS.

## 3. Results

### 3.1. Comparison of Three DNA Extraction Methods

The quantity, purity, and quality of DNA, as well as bacterial composition of ten healthy volunteers’ feces, differed substantially, depending on the method used (Table 1 and Figure 1). The highest DNA yield was obtained with the G’NOME-based method, which resulted in a 252.01 ± 44.67 ng of DNA per mg of feces. This was 1.8 to 40 times higher than the amounts provided by the PROMEGA-based and QIAGEN-based methods, which yielded 139.39 ± 24.65 and 5.93 ± 1.83 ng/mg of feces, respectively (Figure 1a).

Extracted DNA purity patterns, as assessed by the A260/A280 OD ratio (optimal ratio comprised between 1.80 and 2.00), were similar (no statistically significant difference, Kuskall–Wallis test, *p* = 0.40) in the three methods used here (Table 1). DNA quality patterns were assessed by using agarose gel electrophoresis (Appendix A). According to the gel electrophoresis, the PROMEGA-based method yielded longer DNA fragments (of almost 5 kb or more), whereas the G’NOME and QIAGEN-based methods showed small DNA fragments of 200 pb. This suggests that the PROMEGA-based method although producing lowest degradation of DNA is not more accurate for amplification and sequencing processes.

To explore the presence of inhibitors in the fecal DNA solution samples, we tested different amounts of DNA from 0.1 pg to 10 ng using quantitative polymerase chain reaction (qPCR) (Appendix A). The amplification reactions did not show significantly different inhibitory effects between the three methods for DNA amounts that were close to 10 pg, which are usually used in qPCR assays. However, for more concentrated DNA samples (from 100 pg to 10 ng/reaction), the G’NOME- and QIAGEN-based methods showed a higher specific inhibitory effect than the PROMEGA-based method.

Using qPCR to detect and quantify all bacteria and some dominant genera of the gut microbiota, we found significant differences between the QIAGEN, G’NOME, and PROMEGA methods. The number of 16S rDNA gene copies were expressed in logarithm10 of CFU per gram of feces. The amount of all bacteria for the QIAGEN method, with a total bacteria mean of 8.74 ± 1.70 CFU/g of feces, was lower than G’NOME and PROMEGA methods with means of 10.52 ± 0.70, and 10.51 ± 0.09 CFU/g of feces, respectively (Figure 1b). The G’NOME-based method showed significantly higher efficiency for recovering Bifidobacterium genus and *E. coli* species than the QIAGEN-based method. Similarly, the G’NOME-based method showed higher recovery of the *Bacteroides*/*Prevotella* group than the PROMEGA-based method. The latter was also less efficient than the QIAGEN-based method for recovering *Bacteroides*/*Prevotella* genus and *E. coli* species (Appendix A).

### 3.2. 16S rRNA and Taxonomical Assignments

The DNA extracted by the three different methods from ten healthy individuals’ feces were submitted to 16S rRNA sequencing for microbiota characterization. The QIIME1, MALT, and DADA2 pipelines were used for taxonomic assignments.

Principal coordinates analysis outputs from the three DNA extraction methods differed significantly. The highest deviation was observed with the PROMEGA-based method, which yielded a core taxonomy profile that differed significantly from the QIAGEN- and G’NOME-based methods (*p* < 0.001, Principal Coordinate Analysis with PERMANOVA test) (Figure 1c).

Estimation of dominant genera of gut microbiota using the PROMEGA-based method showed better detection of Gram-positive *Ruminococcus*, *Bifidobacterium*, *Blautia*, and *Dorea* genera than the QIAGEN-based method on the same samples but lower detection of Gram-negative *Bacteroides* and *Prevotella* (Appendix A). Compared to G’NOME, the PROMEGA method proved less efficient for *Bacteroides* detection, but more efficient for *Blautia* genus. These differences, which are linked to the DNA extraction methods, could be similarly noted for subdominant bacteria (Appendix A).

Overall, compared to the G’NOME-based method, the PROMEGA-based DNA extraction method appeared to underestimate substantial proportions of gram-negative bacteria, whereas the QIAGEN-based method appeared to miss various Gram-positive bacteria.

### 3.3. 16S rRNA Taxonomic Assignments

WMS and 16S rRNA datasets from the sub-cohort of 50 control individuals from Henri Mondor hospital (see the French cohort in [9] were downloaded from ENA (access number ERP005534). After quality checks, reads were assigned a taxonomy using four pipelines: mOTUs-v2 for the WMS dataset and MALT + MEGAN6 (referred to as MALT), QIIME1, and DADA2 for the 16S rRNA dataset (see Material and Methods). All taxonomic assignments were summarized at the genus level (Appendix A). For the 16S rRNA dataset, the average mean number of reads per sample was 365,160 ± 205,946. QIIME1 identified 21,462 OTU and 564 genera (mean reads per sample: 200,951 ± 134,898; ~55.03% of all reads used for taxonomic assignments). MALT identified 564 taxa and 351 genera (mean reads per sample: 190,252 ± 139,868; ~52.10% of all reads used for taxonomic assignments). DADA2 identified 3623 amplicon sequence variants (ASV) and 383 genera (mean reads per sample: 217,066 ± 148,294; ~59.44% of all reads used for taxonomic assignments). Regardless of the 16S pipeline used, 40–50% of reads were filtered during the taxonomic assignments.

The average mean number of reads per sample for the WMS dataset was 15,625,002 ± 7,195,800 reads (min: 2,304,390; max: 28,086,026; median: 18,033,238), of which 0.03% (~433,359 reads/sample) did not lead to a taxonomic assignment. Moreover, 22.06% of all genera were filtered, many of which were assigned to the “*unknown*” genera (see Materials and Methods).

Overall, 564 genera were merged and identified by at least one out of the four pipelines (Appendix A). Only 160 genera (28.36%) were identified by mOTU-v2, whereas 404 genera (71.23%) were identified by at least one 16S pipeline (158 genera for MALT, 270 genera for DADA2, and 468 genera for QIIME1).

The Venn diagram in Figure 2 shows the overlap between the genera that were identified by WMS and by the 16S pipelines. Overall, 133 genera (83.12%) were identified by WMS and by at least one 16S pipeline. More precisely, 75 genera (46.87%) were identified by MALT and mOTU-v2, 128 were identified by QIIME1 and mOTU-v2 (80.00%), and 119 (74.37%) were identified by DADA2 and mOTU-v2. QIIME1 and DADA2 displayed higher concordance with mOTU-v2 (80.00 and 74.37%, respectively).

#### 3.3.1. Low Abundance Reads

Many genera exhibited low mean abundance. Genera with mean abundance of less than 1% of all reads accounted for 90% of the total abundance with mOTU, for 91.14% with MALT, for 94.66% with QIIME1, and for 92.64% with DADA2. Genera with a mean abundance of less than 0.5% accounted for 83.75, 87.34, 92.30, and 87.77% of the total abundance, respectively.

#### 3.3.2. Abundant Reads 

Among the 51 genera that corresponded to 98% of the total abundance (Appendix A), 30 genera (58.82%, WMS abundance = 2.17 ± 2.76%) where also identified by the three 16S pipelines, 15 genera (29.41%, WMS abundance = 0.49 ± 0.48%) were identified by two 16S pipelines, and one genus (1.96%, WMS abundance = 0.24%) was identified by only one 16S pipeline (Figure 2). When a genus was identified with low abundance by WMS, the probability of detecting it by any of the 16S RNA pipelines was low. With regards to the 16S-pipeline level for these 51 genera, MALT recovered 30 genera (58.83%), QIIME1 recovered 36 genera (70.58%), and DADA2 recovered 35 genera (68.62%). A total of five genera (9.81%, WMS abundance = 0.56 ± 0.21%) were not identified by any of the 16S pipelines (Figure 2). However, four of these five genera did not refer to an official NCBI taxonomy level (“*g_Enterobacteriaceae*” [family level], “*g_Bacteroidales gen. [C Bacteroides/Porphyromonas/Parabacteroides]*,” “*g_Bacteria gen. incertae sedis*,” “*g_Clostridiales*” [order level]). The other genus was “*g_Mycoplasma*” (abundance = 0.29%).

A total of ten genera were not identified by WMS but were identified by at least one 16S pipeline with an abundance above 0.5% (Appendix A). Three genera were identified by the three 16S pipelines: “*g_Sphingomonas*” (mean abundance: 4.69, 4.05, and 4.97%), “*g_Fusicatenibacter*” (mean abundance: 0.65, 0.88, and 0.67%), and “*g_Lachnospira*” (0.77, 0.76, and 1.63%). A total of three genera were identified by two pipelines: “*g_Pseudobutyrivibrio*” (mean abundance: 0.01 and 1.76%), “*g_Paracoccus*” (mean abundance: 1.77 and 1.96%), and “*g_Romboutsia*” (mean abundance: 0.58 and 0.54%). A total of two genera were only identified by one pipeline (DADA2): “*g_Defluviimonas*” (mean abundance: 2.05%) and “*g_Agathobacter*” (mean abundance: 1.63%). For abundance less than 0.5%, the 16S pipelines did not show concordance (using ICC [31] see Appendix A).

A total of four out of these ten genera (“*g_Romboutsia*,” “*g_Fusicatenibacter*,” “*g_Defluviimonas*,” and “*g_Defluviimonas*”) were not found in the mOTU-v2 database, which illustrates discrepancies between the 16S and WGS assignments. The six other genera belonged to the mOTU-v2 database but were not identified in the WGS datasets.

Figure 3 presents the heatmap of abundances (log2 of-) that was estimated for each genus by the three 16S pipelines, including the 51 genera (most abundant according to WMS). The heatmap focuses on 10 samples to aid visualization. This illustrates good agreement between the three 16S pipelines at the sample level and reveals that differences between samples were well conserved.

## 4. Discussion

The characterization of human microbiota is now extensively used to understand host-environment interactions in various diseases, such as cancer, and metabolic and autoimmune diseases. This goal is achieved using molecular gene technologies, since a great number of bacteria from the microbiota are unculturable. Physicians and scientists expect to identify and quantify bacteria using these tools. However, significant variations in the results cast doubt on the output, including various DNA extraction protocols and taxonomic assignations, as well as quantification using various bioinformatics pipelines.

Here, we have compared three DNA extraction methods prior to sequencing using a group of 10 healthy volunteers. Due to its size and homogeneity, this cohort was not expected to cover the large microbiota diversity through humans but allows comparison of DNA extraction methods for identification of bacteria genera in an occidental, young and healthy individuals. We considered the efficiency, quality, and purity of the DNA extracts, as well as their significant impact on relative bacterial abundances and composition outputs, as assessed by standard bioinformatics pipelines. PROMEGA and QIAGEN are available commercial kits that do not include bead-beating. The G’NOME method, which includes a bead-beating step, allows large recovery of microbiota including both aerobic and anaerobic Gram-positive as well as Gram-negatives. Then, 16S RNA- and Whole Metagenomic sequencing (G’NOME-based fecal DNA extracts) from 50 control individuals meaning without disease out of our “French cohort” were considered to identify the most relevant method for 16S RNA taxonomical assignments, using three pipelines. We have shown that 16S RNA-based analyses are mostly concordant with WMS when highly abundant genera were considered, while results regarding low abundant genera (<0.5%) generated significant discrepancies that require additional validation procedures for identification and quantification.

In the present study, the G’NOME method, which is a previously non-automated procedure that includes a bead-beater step to recover DNA from all bacteria [9,15,32], yielded higher purity and lower inhibitory component in samples compared to quicker commercial kit-based methods. The bead-beating step is more accurate than the chemical step (which is solely included in commercial kits) in terms of the destruction of the bacteria wall. This also yields the highest concentrations of microbial DNA, as has been suggested elsewhere [33]. Furthermore, the bead-beating step optimizes the full recovery of DNA material that is suitable for quantification by qPCR.

The second part of our investigation concerned bio-informatics analysis that was performed on 16S rRNA and WMS sequences, which were all obtained from a previous study in which the standard DNA extraction protocol was used [9,18,32]. For the identification and quantification of genera, we used three different 16S rRNA pipelines (MALT, QIIME1, and DADA2), all of which are routinely used by researchers. Then, we compared their outputs to those obtained using the WMS sequences. Taken together, 16S pipelines yielded globally reliable taxonomic assignments. QIIME1 and DADA2 yielded results that were more congruent with WMS taxonomic assignments than MALT. QIIME1 is now outdated. However, it was important to include this version because a vast amount of literature has reported assignations with this version, and we think it is important to provide the community some information concerning its reliability. The updated version, QIIME2, also implements the DADA2 denoising algorithm. We have shown that DADA2 (or its QIIME2 version) is suitable for taxonomic assignment of genera above 0.5%. Moreover, we have also shown that concordance between 16S RNA pipelines and WMS outputs is dependent on the abundance of the genus. Therefore, for low abundant bacteria (<0.5%), for which no WMS reference is available, we advocate the use of a confirmation pipeline, such as WMS, or qPCR.

At the genus level and using the same 16S rRNA database (SILVA, version 128) [34], we found that, for taxonomic assignments, 16S RNA pipelines were mostly concordant with WMS for abundant genera. We found that some genera recovered by the WMS datasets were not recovered by 16S rRNA taxonomic assignment. Moreover, for genera whose abundances were lower than 0.5%, discrepancies were observed between WMS and 16S RNA pipelines. The lower the abundance of a genus (as identified by WMS), the lower the probability of it being correctly detected by any of the three 16S RNA pipelines. These results suggest a need for caution when extrapolating the relevance of bacteria identification by 16S RNA analysis for low abundant genera. Bias could be reduced by using two 16S pipelines for the identification and/or qPCR quantification. This latter point is of the utmost relevance since genus/species abundances are incorporated into complex statistical models to identify relevant biomarkers without additional verification. In addition, many studies focus on the fold changes in genera abundance between subgroups without reporting the mean abundance of genera and without assessing the quality of this estimation.

For the OTU approach, recent literature questions the canonical threshold of 97%, which is currently used, challenging the reliability for species identification [35,36]. Here in the absence of a “Gold standard” we did not judge that tweaking the OTU pipeline was needed since we limited the analysis to the genus level.

We estimated that abundances lower than 0.5% of all bacteria as reported by a 16S pipeline yield higher biases in the estimation of the bacteria (here genus) assignation and quantification. Thus, 0.5% can be considered the threshold below which 16S RNA assignations could be doubted, which would subsequently require additional confirmation.

Alternatively, these discrepancies might be a consequence of “multi-mapping” reads, as such reads could match several (highly similar) sequences that enable different taxonomies. For example, MALT uses the lowest common ancestor (LCA) algorithm, which is implemented in MEGAN6 to displace reads to a lower taxonomic level (i.e., “family” or “order”) when they are not identified at the genus level. Conversely, QIIME1 and DADA2 cluster these reads and only pick one of all the possible taxonomic assignments. Nevertheless, the estimated abundances in each sample, at the genus level and for the three 16S rRNA pipelines, present an acceptable taxonomic level for discussing concordance among the pipelines. 

Taken together, our results show that bacteria identification (genus assignment) and abundance (percentage of reads per genus) based on 16S RNA sequencing exhibit good concordance with WMS references for abundant genera. Whereas concordance between 16S RNA and WMS decreases for low abundance genera, the same conclusion can be drawn when focusing on genera that is only identified by 16S rRNA. Considering these results, we suggest a threshold of 0.5% of the total abundance to define “low” abundance. Below this threshold, we suggest the need of two bioinformatics pipelines, qPCR, and/or WMS, to confirm and validate the identification and quantification of bacteria.

## 5. Conclusions

Molecular characterization of microbiota is a good alternative to cultures-based methods, which can be applied to clinical and translational trials using 16S sequencing. However, this should be limited to genus abundance estimations. The choice of DNA extraction method is crucial for wide bacterial genera detection, and we recommend the G’NOME-based method. Diagnosis and quantification of high abundant bacteria may be achieved using 16S rRNA sequencing, using the DADA2 approach as reliable pipeline. The limitation concerns low abundant bacteria for which two pipelines are mandatory (estimated threshold set at <0.5%), and WMS should be required to validate findings. Alternatively, qPCR may be necessary also to validate the relative quantification of various bacteria through communities.

## Figures and Tables

**Figure 1 microorganisms-08-01954-f001:**
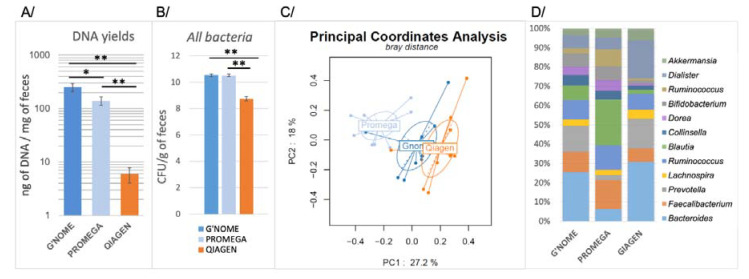
Microbiota characterization following three fecal DNA extraction methods. Comparison between G’NOME, a standard reference DNA extraction method and two (PROMEGA and QIAGEN) routine calibrated methods from feces of 10 healthy volunteers. Fresh fecal samples immediately submitted to DNA extraction. Comparisons between methods were performed by using Kruskall–Wallis and Wilcoxon rank test with * *p* < 0.05, ** *p* < 0.001. (**A**) DNA yields were calculated with Qubit^®^ reading as follow: total DNA quantity obtained for extraction of one milligram of fresh feces. (**B**) Levels of *all bacteria* were assessed using a single bacterial qPCR detection technique and expressed as Log10 of CFU/g of feces (means ± SEM). (**C**) Principal coordinate analysis based on the 16S RNA sequences on genus levels. PC1 and PC2 axes represent 27.2% and 18.0% of the variance within the microbial community. (**D**) Relative abundance of 12 dominant genera.

**Figure 2 microorganisms-08-01954-f002:**
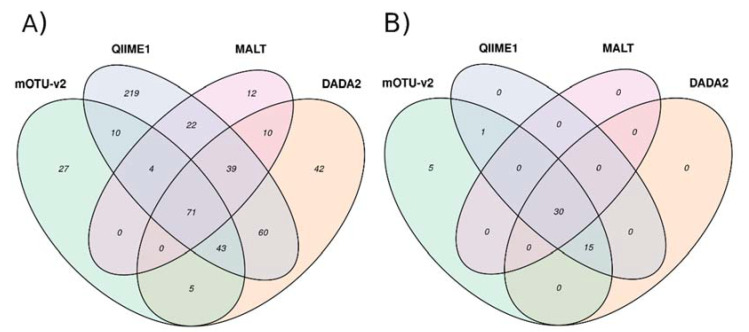
VENN diagram of overlapping genera identified by the four pipelines. (**A**) Overlapping of the 564 genera. (**B**) Overlapping of the 51 most abundant genera according to WGS abundances estimation.

**Figure 3 microorganisms-08-01954-f003:**
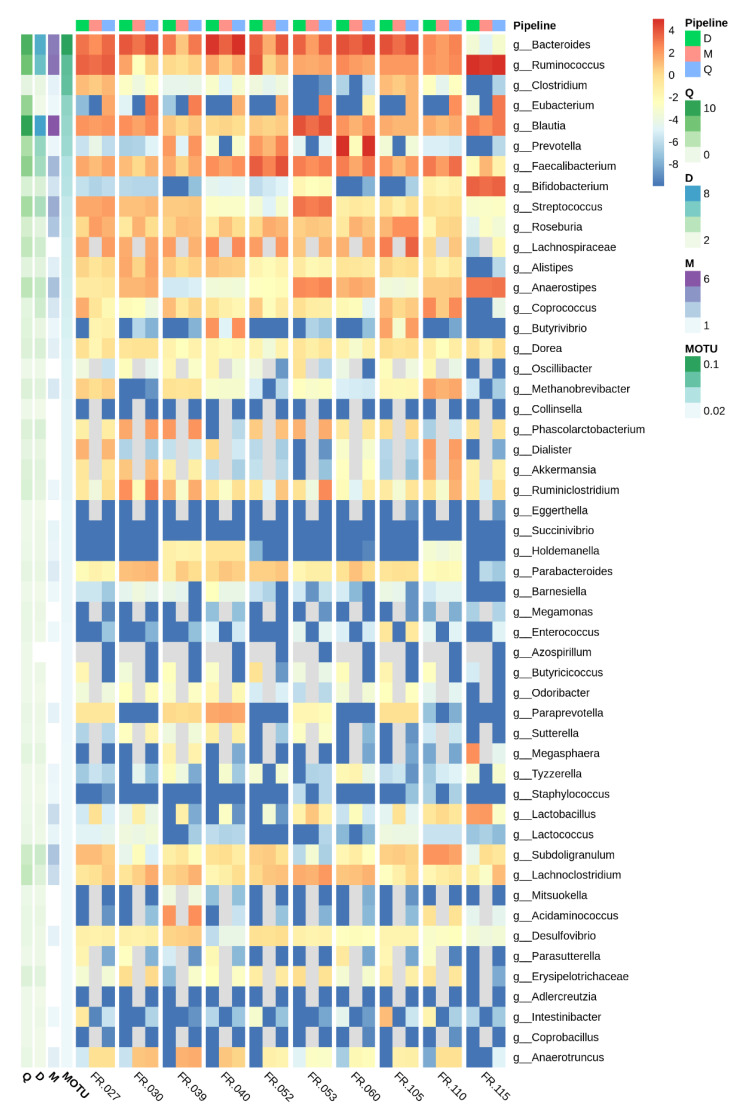
Heatmap of genera abundances at the sample level estimated by the three 16S rRNA pipelines, for the 51 most WMS abundant genera. Left annotation of the heatmap: average over the 50 samples estimated by each pipeline. Abundances are represented as log2 (percentage). Genus abundances estimated by each pipeline and for 10 samples for visual convenience.

**Table 1 microorganisms-08-01954-t001:** Main characteristics of three DNA extraction protocols. Synthesized advantages and disadvantages of DNA extraction methods. L: lysozyme, T: temperature, CLB: cell lysis buffer, BB: bead beating, M: mutanolysin.

DNA Extraction Method	Kits and References	Company	Lysis Procedure	Handling Time	DNA Yields (ng/mg)	DNA purity (A260/A280)
**G’NOME**	G’NOME DNA isolation kit^®^ (#112010600)	MP Biomedicals Santa Ana, CA, USA	BB, CLB, T	24h	252.01 ± 44.67	1.74 ± 0.02
**PROMEGA**	Wizard Genomic DNA purification kit^®^ (#A1120)	Promega Madison, WI, USA	L, M, CLB, T	7h	139.39 ± 24.65	1.69 ± 0.04
**QIAGEN**	QIAamp DNA Stool Mini Kit^®^ (#12830)	Qiagen Hilden, Germany	CLB, T	1h	5.93 ± 1.83	2.20 ± 0.06

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
