# Peer review of "The Limits and Avoidance of Biases in Metagenomic Analyses of Human Fecal Microbiota"

_microorganisms, 2020, doi:10.3390/microorganisms8121954_

Round 1
Reviewer 1 Report
Dear Dr. Colomer Lluch and authors,
this article represents a lot of work and time to put such a big article together. Félicitations!
The research presented in this manuscript is very interesting and relevant. I particularly appreciated the quality of the language and vocabulary used throughout this article because it made the reading easier despite how complicated the work is. I suggested very little edits.
I truly believe this article can reach up to a publication level once additional clarifications and work are done. Particularly the presentation of the methodology needs to be improved, and the way results are reported as well. The discussion also needs to cover a few more aspects, and to go deeper for some of the current ones.
I think authors have most of the data they need already but, they need to rework the way they present it (visually, textually, etc.). I it is hard to see "outside of the box" and having to explain concepts that appear obvious to authors but, it was not clear for me in general. Particularly, the sequencing part was cryptic in my opinion. I didn'T understand what was going on with a dataset previously prepared by another group (Milanes et al.). I also was very puzzled by how they could quantify, by qPCR, an E. coli species using SYBR-green rather than a TaqMan probe. This affected the way I could fairly assess the article. I suggested using an R package to deal with their incertae sedis issues.
I am attaching an annotated PDF version of the article with all my comments, edits and suggestions. I integrated comments for the supplementary files into this PDF as well. I hope this review will guide the authors to improve their work and publish the tremendous work they have done.
Good luck.

Reviewer 2 Report
Major concerns:
- When the manuscript’s topic is on biases in analyses, the authors were biased in methods, at least in DNA extraction methods. The authors recognized G’NOME method as the gold standard at the very beginning, but little evidence to persuade readers: though higher in yield, lower in quality (OD260/280) than Qiagen’s, more fragmentation than Promega’s, more expensive, more turnaround time (so obviously), and less compatible to automation.
- Details of sequencing (16S and WGS) protocol are missing in materials and methods, in a comparison to other parts. 16S V3-V4 region is variable so as to be used for taxonomy classification, not conserved, though the primers used for PCR are conserved. CCR1 cohort? WGS? DNA for WGS? Sequencing is the key to the following bioinformatics analysis.
- While supplementary tables and figures were unavailable to me, I think a (supplementary) table on sequencing data, qPCR data and bioinformatics data regarding each sample will be usable and helpful for readers, especially the variations among samples could be demonstrated and clarified in comparisons.
- Database is a key to bioinformatics analysis of taxonomy classification, which should have been discussed, especially the current databases available are not comprehensive. It is regretted that authors used only SILVA for 16S and mOTU-v2 for WGS and that the two databases seemed not compatible in direct comparisons. More databases or more 16S/WGS pipelines will be appreciated.
Minors:
- Sentence in Line 54 needs revision;
- SRA annotation in Line 22 should be in Line 166;
- Sentence in Line 215 need revision;
- MOTU in Line 310 should be mOTU?
Round 2
Reviewer 1 Report
Dear Dr. Colomer Lluch and authors,
This revised version of the article has improved significantly. Félicitations!
I suggested a few edits in the text, nothing major.
I am attaching an annotated PDF version of the article editing comments.
The presentation of the methodology was much improved, especially the sequencing part which is quite clear now. Authors properly named the type of sequencing they preformed (WMS as opposed to WGS) and further explained their 16S protocol. Explanation of the “The French cohort” dataset they reused from Zeller et al. is now clearly presented. As I requested, authors also clarified how they could use SYBR-green to quantify E. coli (with a melting curve). Authors really took the majority of my suggestions into account and applied them beautifully.
The presentation of results overall is also very well improved. I was however disappointed that the “incertae sedis” problem previously raised was simply acknowledged, as authors responded that “they would study it more closely” but, I didn’t see modifications in the article accordingly. Specifically, I had mentioned that authors could treat that with an R package I pointed for them, along with the name of the function they would have to run. I understand that this represent additional work but, it would possibly affect their diversity analyses and allow them to assign taxonomic levels more accurately. Sometimes, incertae sedis is assigned to the family or order levels but, the genus and species are still properly identified. This can be important for taxonomic assignments. I can imagine that they were time-crunched for their review and could not deliver on time.
Perhaps in the previous round some of my comments were not clear so, I would just like to attempt re-explaining my comment on their usage of “Real-time quantitative PCR” expression given that authors responded that it was both (quantitative and real-time). As far as I know, it is indeed both, but real-time and quantitative PCR are “synonyms”, hence my suggestion to pick one or the other in the previous round (i.e. either quantitative PCR or real-time PCR). I found a generic definition of qPCR and extracted the beginning of it here: “A real-time polymerase chain reaction (real-time PCR), also known as quantitative Polymerase Chain Reaction (qPCR), is […]” that may help them understand my point.
I was also surprised that after my concern over their “consideration for diversity” (of patients studied, because they used data from a group of caucasian young healthy males), they simply removed that information. There is likely a bias in their diversity analyses caused by this limited group of patients. Although it is ok, authors shall recognize that in their discussion.
I was excited to see that authors shared their raw data and their in-house scripts on GitHub. They also added more information on the packages/tools used in their methodology but, a few are still missing. I highlighted those in the annotated PDF text.
I was not satisfied with the authors response to my comment on their ability to address unexpected taxonomic discrepancies in their data: “This has been a surprising point for us because the three pipelines used the same database and we expected that the nomenclature should be the same. We now have this experience to tweak our pipeline”. I think they should therefore tweak their pipeline and modify their output data accordingly for this article. Again, I can imagine that they were time-crunched for their review.
I still don’t like that one of their figure is “duplicated” in the article: Figure 3 is a “crop” of Supplementary Figure S5. Typically, research articles shall avoid at all cost to repeat information. As well, in the previous round of review I requested them to remove legends to accompany figures captions but, I still see them in the supplementary files.
In the previous review round, I mentioned to authors: “Throughout the article, can authors ensure they took Error Analysis and Significant Figures into account when reporting error margins (e.g. SEM, percentages, etc.) and that those standards were applied everywhere in the manuscript, please?”. I can see that authors tried to handle my comment that was perhaps not clear since I see it was not fully rectified. Significant figures are the numbers of digits to include or exclude depending on the type of data and their errors. For instance, percentages will have less significant figures carried over than other types of measurements and there are specific rules to follow, depending on the type of data, zeros, periods, etc. Authors shall consider those rules to properly reports their data in the article.
Authors used a 97% cutoff for OTUs but, this is a very controversial assignment value for bacteria. Some scientists think it shall be higher. Authors need to elaborate/acknowledge on that aspect in the discussion.
That is all, again, congratulations on all the work.

Reviewer 2 Report
concerns addressed. manucript accepted
Author Response
no request from this reviewer.